# Neutralizing antibody responses to SARS-CoV-2 in symptomatic COVID-19 is persistent and critical for survival

Stefania Dispinseri [1], Massimiliano Secchi[2,9], Maria Franca Pirillo[3], Monica Tolazzi[1], Martina Borghi[4], Cristina Brigatti[2], Maria Laura De Angelis[5], Marco Baratella[1], Elena Bazzigaluppi[2], Giulietta Venturi [4], Francesca Sironi[1], Andrea Canitano[3], Ilaria Marzinotto[2], Cristina Tresoldi[6], Fabio Ciceri[7,8], Lorenzo Piemonti[2,8], Donatella Negri[4,10], Andrea Cara [3,10], Vito Lampasona[2,10] & Gabriella Scarlatti [1,10✉]

Understanding how antibody responses to SARS-CoV-2 evolve during infection may provide important insight into therapeutic approaches and vaccination for COVID-19. Here we profile the antibody responses of 162 COVID-19 symptomatic patients in the COVID-BioB cohort followed longitudinally for up to eight months from symptom onset to find SARS-CoV-2 neutralization, as well as antibodies either recognizing SARS-CoV-2 spike antigens and nucleoprotein, or specific for S2 antigen of seasonal beta-coronaviruses and hemagglutinin of the H1N1 flu virus. The presence of neutralizing antibodies within the first weeks from symptoms onset correlates with time to a negative swab result ($p = 0.002$), while the lack of neutralizing capacity correlates with an increased risk of a fatal outcome ($p = 0.008$). Neutralizing antibody titers progressively drop after 5–8 weeks but are still detectable up to 8 months in the majority of recovered patients regardless of age or co-morbidities, with IgG to spike antigens providing the best correlate of neutralization. Antibody responses to seasonal coronaviruses are temporarily boosted, and parallel those to SARS-CoV-2 without dampening the specific response or worsening disease progression. Our results thus suggest compromised immune responses to the SARS-CoV-2 spike to be a major trait of COVID-19 patients with critical conditions, and thereby inform on the planning of COVID-19 patient care and therapy prioritization.

[1] Viral Evolution and Transmission Unit, IRCCS Ospedale San Raffaele, Milan, Italy. [2] Diabetes Research Institute, IRCCS Ospedale San Raffaele, Milan, Italy. [3] National Center for Global Health, Istituto Superiore di Sanità, Rome, Italy. [4] Department of Infectious Diseases, Istituto Superiore di Sanità, Rome, Italy. [5] Department of Oncology and Molecular Medicine, Istituto Superiore di Sanità, Rome, Italy. [6] Molecular Hematology Unit, IRCCS Ospedale San Raffaele, Milan, Italy. [7] Hematology and Bone Marrow Transplantation Unit, IRCCS Ospedale San Raffaele, Milan, Italy. [8] School of Medicine and Surgery, Università Vita-Salute San Raffaele, Milan, Italy. [9] Present address: DNA Enzymology & Molecular Virology Unit, Institute of Molecular Genetics, National Research Council, Pavia, Italy. [10] These authors jointly supervised this work: Donatella Negri, Andrea Cara, Vito Lampasona, Gabriella Scarlatti. ✉email: scarlatti.gabriella@hsr.it

Severe acute respiratory syndrome coronavirus 2 (SARS-CoV-2) infection can be fatal in a significant proportion of hospitalized Corona Virus Disease 19 (COVID-19) patients despite the development of an anti-viral antibody response[1]. Studies in large cohorts of SARS-CoV-2 infected individuals indicate that antibodies to the receptor-binding domain (RBD) of the viral spike glycoprotein appear within the first three weeks from symptoms onset and that IgG and/or IgA seroconversion occurs either sequentially or simultaneously with the appearance of IgM[2,3]. Using a newly developed Luciferase Immunoprecipitation System (LIPS) assay, we previously detected anti-RBD IgG in 95% of COVID-19 patients by the fourth week from symptom onset and observed that these antibodies increased throughout follow-up until the third month post-hospital discharge[4]. Moreover, the early presence of anti-RBD IgG and anti-spike IgA positively correlated with patient survival and reduced persistence of SARS-CoV-2 RNA in naso-pharyngeal swabs, respectively.

The spike glycoprotein mediates entry into target cells via the ACE2 receptor and clinical trials with monoclonal antibodies (mAbs) against its RBD decreased viral load in patients with recently diagnosed mild/moderate COVID-19[5–7]. Furthermore, anti-spike neutralizing antibodies (nAbs) produced by COVID-19 patients can block viral infection of human cells in vitro and counter viral replication in vivo[3,6,8,9]. However, the impact of nAbs on COVID-19 course is still controversial with some studies even suggesting either a detrimental role for nAbs in disease progression[8,9] or finding no nAbs differences among hospitalized patients who subsequently experienced varying disease outcomes[3].

In addition, studies dealing with infections carried out by MERS and SARS-CoV beta coronaviruses, two viruses closely related to SARS-CoV-2, suggest that the anti-viral humoral immunity, sometimes still detectable more than a year after hospitalization, wanes over time[10,11]. Whether SARS-CoV-2 nAbs decline at a similar pace is yet to be conclusively established. In light of the current timing of the pandemic, most published serological studies are predominantly cross-sectional or at most include a longitudinal follow-up of few months[3,12–14].

Here we provide a comprehensive antibody profile of well-characterized COVID-19 symptomatic patients followed longitudinally for up to eight months from symptom onset. We test 162 patients using a newly developed, robust lentiviral vector (LV)-based SARS-CoV-2 neutralization assay and the LIPS assay for the detection of IgG, IgM, and IgA to SARS-CoV-2 spike, RBD and nucleoprotein (NP) as well as IgG to the spike S2 protein of seasonal beta coronaviruses, and to H1N1 influenza virus hemagglutinin. Our results indicate that early development of nAbs is critical for patient survival and virus control, supporting the early introduction of mAb therapy after infection in selected severely ill patients, and suggest that nAbs induced by prophylactic vaccination will most likely be long-lasting.

## Results

**COVID-BioB study cohort characteristics**. We profiled the humoral immune response of 162 patients within our COVID-BioB cohort. All had a confirmed SARS-CoV-2 infection and a record of symptoms onset. Serum samples were collected at hospital admission in March-April and at the outpatient visits during follow-up until Nov 25, 2020 (Table 1). Patients were predominantly White Europeans (89.5%), male (66.7%) and had a median age of 63 years. A relevant fraction had a body mass index >30 (32.4%) and 57.4% had one or more co-morbidity, with hypertension (44.4%) and diabetes (24.7%) being the most frequent (Table 2). Patients sought an emergency hospital visit after a median of nine days from symptoms onset and 134 were hospitalized. At admission, symptoms consisted prevalently of fever, fatigue, and respiratory difficulties often associated with dyspnea. The median hospitalization duration was 14 days (95%CI 7–24). Overall, 26 patients entered the Intensive Care Unit (ICU) and 29 passed away, of whom 11 after ICU entry. The median time to a negative RT-PCR result of the naso-pharyngeal swab was 40 days (95%CI 37–43). Patients who recovered were followed-up for a median of 201 days (range 145–250, 95%CI 196–205) from symptom onset. The set of laboratory blood tests recorded according to the COVID-BioB protocol is presented in Supplementary Table 1. The general characteristics of this study cohort reflect those of the overall COVID-19 patients admitted to our hospital during the same period and previously described[4,15].

**Truncated Spike on LV pseudovirions improves infectivity and is suitable for SARS-CoV-2 neutralization assay**. To test the nAb response we developed and optimized a Lentiviral (LV)-based SARS-CoV-2 neutralization assay, that does not rely on ACE2 receptor-expressing engineered cell lines and is suitable for handling in a Biosafety Level 2 environment. This assay is based on Luciferase expressing LV (LV-Luc) pseudoparticles, enveloped with a spike glycoprotein containing a 21aa deletion in the cytoplasmic tail (Spike-C3) (Supplementary Fig. 1). This truncation removes an endoplasmic reticulum (ER) retention signal that interferes with the budding of viral particles[16,17], thus favoring the interaction of spike and Gag proteins during particle assembly[18]. Upon transfection into 293 T Lenti-X cells of either

**Table 1 Clinical management and serum samples availability of 162 COVID-19 study patients.**

| Clinical management | No. of patients | No. of patients with serum samples at visit performed | |
|---|---|---|---|
| | | In-hospital | Outpatient clinic |
| Admitted to the Emergency Department | 150 (92.6%) | | |
| Discharged (not hospitalized) | 16 (10.7%) | 16 | 9 |
| Hospitalized | 134 (89.3%) | | |
| ≤7days | 25 (18.7%) | 25 | 19 |
| >7days | 65 (48.5%) | 65 | 44 |
| deceased | 18 (13.4%) | 18 | na |
| in need of ICU, recovered | 15 (11.2%) | 15 | 2 |
| in need of ICU, decease | 11 (8.2%) | 11 | na |
| Admitted at the COVID-19 outpatient clinic | 12 (7.4%) | | |
| Not hospitalized | 10 | na | 3 |
| Hospitalized | 2 | na | 2 |

ICU Intensive Care Unit, na not applicable
A dedicated serum sample was collected during attendance at the hospital, which took place either at the Emergency Department or at the ward (called the in-hospital sample). Follow-up serum samples were collected at the COVID-19 outpatient clinic during the 1, 3 and 6 months visits after discharge

**Table 2 Characteristics of the COVID-19 study population (No. = 162).**

| Characteristics | COVID-19 patients | Missing data |
|---|---|---|
| Age, years (median, 95%CI) | 63 (52–72.5) | 0 |
| Sex Male | 108 (66.7%) | 0 |
| Ethnicity | | 0 |
| White Europeans | 145 9.5%) | |
| Hispanic | 12 7.4%) | |
| Asian | 4 (2.5%) | |
| African | 1 (0.6%) | |
| Co-morbidities | | 0 |
| Hypertension | 72 (44.4%) | |
| Coronary artery diseases (CAD) | 21 (13%) | |
| Diabetes | 40 (24.7%) | |
| Chronic obstructive pulmonary disease (COPD) | 6 (3.7%) | |
| Chronic Kidney Disease (CKD) | 24 (14.8%) | |
| Cancer | 17 (10.5%) | |
| Neurodegenerative disease (ND) | 5 (3.1%) | |
| Number of co-morbidities | | 0 |
| None | 69 (42.6%) | |
| 1 | 29 (17.9%) | |
| 2 | 42 (25.6%) | |
| 3 or more | 22 (13.6%) | |
| Body mass index (BMI) | | 14 |
| <25 | 40 (27%) | |
| 25–30 | 60 (40.5%) | |
| >30 | 48 (32.4%) | |
| Symptoms at disease onset | | 5 |
| General | | |
| Fever | 84.1% | |
| Headache | 24.2% | |
| fatigue/malaise | 56.7% | |
| myalgia/arthralgia | 33.1% | |
| Respiratory | | |
| Cough | 61.1% | |
| Dyspnea | 65.6% | |
| Sore throat | 17.8% | |
| Chest pain | 22.9% | |
| Gastrointestinal | | |
| Diarrhea | 28% | |
| Vomiting/nausea | 14% | |
| Abdominal pain | 8.9% | |
| Others | | |
| Conjunctivitis | 14% | |
| Hypo/anosmia | 35.7% | |
| Hypo/dysgeusia | 39.7% | |
| Skin rash | 6% | |
| Median days (95%CI) from symptoms onset: | | 0 |
| To hospital admission | 9 (5–12) | |
| To first blood sampling for Biobank | 11.5 (7–18) | |
| Median days of hospital stay of 134 hospitalized patients (95%CI) | 14 (7–24) | 0 |
| Median days from symptoms to negative RT-PCR swab (95%CI) | 40 (37–43) | 7 |
| Median follow-up days (95%CI) | 201 (196–205) | na |

*na* not applicable
If not otherwise defined, the number of patients for each listed characteristic is reported. The median BMI was 27.84 (IQR: 24.48-31.42). Missing data are indicated for each group of categories

full-length spike or Spike-C3 expression plasmids, both constructs showed comparable expression levels on the cell surface (Fig. 1a). Conversely Spike-C3 was incorporated on the pseudoparticles' surface at higher levels than the whole spike, though Gag protein content was comparable across all preparations (Fig. 1b).

LV-Luc/Spike and LV-Luc/Spike-C3 infectivity were tested in a panel of four ACE2 expressing epithelial cell lines Caco2 (human, colon), Calu3 (human, lung), HUH7 (human, liver), VEROE6 (African green monkeys, kidney), as well as one ACE2-negative human B cell line, Raji. Based on the growth and cell confluence, 20.000 VEROE6, 70.000 Caco2, 50.000 Calu3, 7.000 HUH7, and 70.000 Raji cells per well in 96-well plates were used for the infectivity experiments to be evaluated for luminescence at 48 h,

with increasing dilutions of the LVs normalized for RT-activity. Due to the higher incorporation of Spike-C3 on the pseudoparticle surface, LV-Luc/Spike-C3 was consistently more infectious than LV-Luc/Spike, with an infection capacity three to 50 times higher depending on the cell line used (Fig. 1c). The difference was more pronounced in VEROE6 cells (50-fold), followed by HUH7 cells (9-fold) and Caco2 (3-fold), while in Calu3 cells only Luc/Spike-C3 showed some low-level infection (maximum 250 relative luciferase units (RLU)). As expected Raji cells were not infected. The assay background level did not exceed 100 RLU for all the cell lines tested (Fig. 1c). Thus, for the development of the neutralization assay the LV-Luc/Spike-C3 particles and the VEROE6 monkey epithelial kidney cell line, which showed high ACE2 expression (Supplementary Fig. 2), were selected. The optimization of key neutralization assay parameters is described in Methods.

**Antibody responses to SARS-CoV-2 Spike appear early after symptoms onset, though not in the totality of patients.** Profiling of 162 COVID-19 patients' sera was performed through the SARS-CoV-2 nAb assay and the LIPS binding assay towards SARS-CoV-2 spike domains (S1 + S2, S2, RBD) and NP, the spike S2 of HCoV-OC43 and HCoV-HKU1 and the HA1 protein of the 2009 H1N1 flu virus. Sera from 150 patients during the hospital stay were available, with the sampling occurring between 1–38 days from symptom onset. Early nAb responses displayed a wide heterogeneity in magnitude (median ID50 1/786, range 1/<40-1/1,660,000) (Fig. 2a and Supplementary Table 2). NAbs were present in 43.2% patients (19 of 44, median ID50 1/410) sampled within the first week and in 78.9% (45 of 57, median ID50 1/1165) within the second week from symptom onset, respectively. Overall, 28.7% (43 of 150) of patients had no detectable nAbs in their in-hospital sample. The appearance of IgG to spike proteins was similar to that of nAbs (Fig. 2b, c, d). Indeed, the best binding antibody correlate of nAbs were IgG to any of the spike antigens used in LIPS (Fig. 3b, c, and Supplementary Fig. 3), Spike IgG thus behaving as a good surrogate marker for nAbs detected in our neutralization assay. IgM and IgA binding to the spike proteins were more frequently detected than IgG during the first two weeks from symptoms onset (Fig. 3a); then, as expected, they progressively declined during follow-up (Fig. 3b,c). Spike and NP antibodies showed similar frequencies in samples taken 3-4 weeks post-symptom onset.

**Kinetics and persistence of nAbs to SARS-CoV-2 during longitudinal follow-up.** NAbs were still rising in 71.6% of the 87 patients sampled (median ID50 1/3190; range 1/<40–161,000) at the first outpatient's visit at a median 42 days from symptoms onset (range 22–90), whereas at the second visit (median 99 days, range 62–138) nAbs declined in all 77 tested patients (median ID50 1/909, range 1/<40–28,000) (Fig. 2a and Supplementary Table 2), with a complete loss of nAbs in three patients. Forty-six patients attended additional visits at or beyond 6 months (median 204 days, range 145–250) and their nAb titers were further decreasing (median ID50 1/660, range 1/<40–10,900), but no additional patient lost the nAb response. Overall, 67.3% (101 of 150) COVID-19 patients developed nAbs rapidly within 2 weeks from symptoms onset, which persisted up to 36 weeks in all but three patients.

The kinetics of IgG response to spike proteins reflected that of nAbs (Supplementary Table 2): antibodies to S1 + S2 (Fig. 2b) reached a peak by 5-8 weeks and then declined, while those to RBD (Fig. 2c) and S2 (Fig. 2d) continued to rise throughout follow-up, but still correlated significantly with nAbs (Fig. 3 and Supplementary Fig. 3).

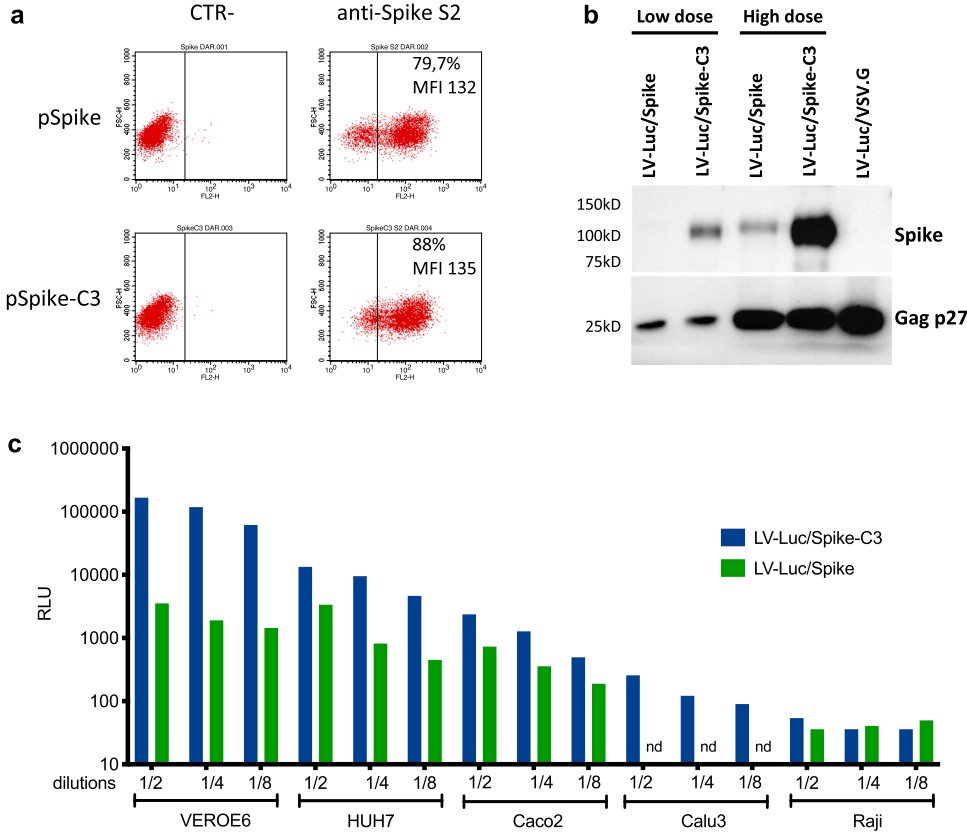

**Fig. 1 Higher incorporation of truncated Spike on LV pseudovirions improves infectivity in ACE2 + cell lines. a** Evaluation of SARS-CoV-2 spike expression in 293T Lenti-X cells transfected with pSpike and pSpike-C3 plasmid by flow cytometry. Data from one representative experiment of three independent experiments are shown. **b** Western blot (WB) of lysates from concentrated preparations of LV-Luc pseudotyped with wild-type spike (LV-Luc/Spike) or spike truncated (21aa) at the cytoplasmic tail (LV-Luc/Spike-C3). Low dose: $1 \times 10^5$ reverse transcriptase (RT) units; High dose: $6 \times 10^5$ RT Units. Control vector included LV-Luc pseudotyped with VSV.G envelope (LV-Luc/VSV.G). Data from one representative of three independent experiments are shown. **c** Infection of human and macaque epithelial cell lines and the control B cell line Raji incubated with decreasing concentrations of LV-Luc pseudotyping the full length (LV-Luc/Spike) or the truncated form (LV-Luc/Spike-C3) of the SARS-CoV-2 spike. Data are expressed as mean luciferase activity (RLU) of duplicates. Nd = not done. Source data are provided as a Source Data file.

As previously reported, antibody persistence may vary according to the degree of the disease severity[19]. In our study all patients were symptomatic at disease onset, though 26 did not require hospitalization (Supplementary Table 3). Their characteristics were different from the overall cohort, with the majority being younger females (61.5%, median age 49.5 years) without co-morbidities (80.8%), although with symptoms at disease onset similar to hospitalized patients, except for dyspnea. In non-hospitalized patients, the profile of nAbs and binding antibodies to spike proteins was similar to that of hospitalized patients (Supplementary Fig. 4). None of the 12 non-hospitalized patients tested during follow-up lost the nAb response until the 2nd and 3rd outpatient visit (range 76–218 days after symptoms onset).

**Development of neutralizing antibody after symptoms onset correlates with viral control.** Prevention of a critical COVID-19 illness is a major hurdle and biomarkers are strongly needed to drive the choice of therapeutic interventions. We performed an extensive analysis of antibody responses, clinical and laboratory parameters collected at or in close proximity of the in-hospital sampling putting them in relation to time to a negative RT-PCR result of the naso-pharyngeal swab and to patients' survival. In the univariate analysis the presence of nAbs was the only other positive strong predictor of time to a negative swab test (HR 2.238; 95%CI 1.343–3.734), besides IgA to SARS-CoV-2 S1 + S2 (HR 1.958; 95%CI 1.079–3.554) (Supplementary Fig. 5). Early development of nAbs significantly correlated with the time to a negative swab (p = 0.002), i.e. clearance of the virus, when stratified for time from symptoms onset (Fig. 4a). It was instead impossible to investigate the correlation with viral load at disease onset since a quantitative RT-PCR assay was not yet available when swabs were analyzed.

**Lack of a neutralizing antibody response predicts fatal outcome.** NAbs may limit viral spread, and consequent disease progression, when developed early after infection. In the 134 hospitalized patients nAbs detected at the time of the in-hospital sampling did not show any impact on the hospitalization length (Supplementary Table 4), but significantly correlated with critical illness (Supplementary Fig. 5). Importantly, lack of a nAb response early after infection was a strong predictor of death when stratified for time from symptoms and corrected for age and sex (HR 2.918, 95%CI 1.321–6.449; p = 0.008) (Fig. 4b). This correlation was still maintained when a more stringent ID75 for the nAb response was applied (HR = 2.67, 95%CI 1.2–5.9; p = 0.015). In addition, the correlation was further maintained when corrected for each one of the identified correlates of death, including co-morbidities and relevant immune-inflammatory markers, and the absence of IgA to S1 + S2 (Supplementary Fig. 6).

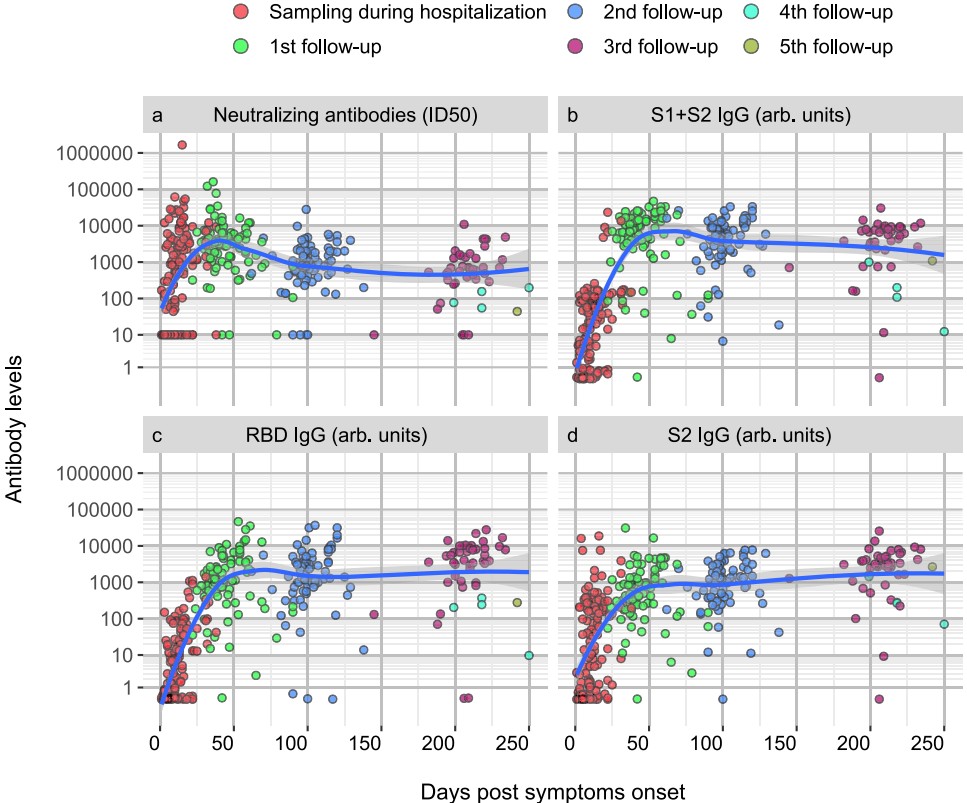

**Fig. 2 Kinetics of the anti-Spike antibody response.** Scatterplots of each patient anti-spike neutralizing (**a**), S1 + S2 IgG (**b**), RDB IgG (**c**), S2 (**d**) IgG antibodies over time from symptoms onset. Antibody levels correspond to the reciprocal of the ID50 for nAbs or to arbitrary units for all other reactivities. Sampling during hospital attendance (ER or ward) or at post-discharge outpatient follow-up visits is shown by the indicated color code. The moving average of data + SE (black curve line + gray band), as obtained by a LOESS curve fitting polynomial regression, is displayed. Source data are provided as a Source Data file.

**Kinetics and persistence of neutralizing antibodies is preserved despite advanced age and co-morbidities.** The impact of advanced age and diverse co-morbidities on nAb response kinetics was investigated (Supplementary Table 4). Low or no nAbs in the first weeks after symptoms onset (Fig. 5) were detected in those patients above 63 years of age, or presenting one or more co-morbidities, who passed away. Patients without an early nAb response were significantly older with debilitating co-morbidities, including cancer or chronic pulmonary and kidney diseases. Noteworthy, these patients presented frequently a disease onset with less severe COVID-related respiratory symptoms that did not require ICU admission, and clinical laboratory tests mostly showing a lower immune-inflammatory activation with organ involvement. Conversely, in patients who recovered, advanced age and presence of co-morbidities did not impact on the distribution of early antibody response and, analogously, titer, kinetics and persistence of nAbs after recovery were not affected by either parameter.

**Antibody responses to seasonal coronaviruses are temporarily boosted without worsening disease progression.** Pre-existing antibody responses to other beta coronaviruses have been proposed to potentially have either a detrimental or beneficial role in COVID-19[20,21]. Testing of IgG to S2 of HCoV-OC43 and -HKU1 in samples collected during the first weeks from symptoms onset did not correlate with either time to a negative swab test or death (Supplementary Fig. 5). However, in the same time window, IgG titers to S2 of HKU1 and OC43 correlated with SARS-CoV-2 nAb titers (Fig. 6a, and Supplementary Fig. 7). Interestingly, patients who developed nAbs to SARS-CoV-2 during the first three weeks

from symptoms onset had significantly higher IgG titers to either HCoVs compared to patients with undetectable nAbs, a difference that persisted throughout follow-up (Fig. 6b, c). Conversely, a pre-existing or ongoing Flu infection, as documented by an antibody (IgM and/or IgG) response to the H1N1 flu virus HA antigen in the in-hospital samples, did not correlate with the presence of SARS-CoV-2 nAbs nor survival (Fig. 3b, and Supplementary Table 2), further indicating that the immune response was not compromised in general.

## Discussion

Two new and robust assays were developed to comprehensively profile the humoral response in symptomatic COVID-19 patients infected during the first wave of the pandemic. Our longitudinal study contributes to the understanding of three important aspects of the SARS-CoV-2 infection, with relevant implications for the future of the pandemic. Firstly, nAbs are a correlate of survival as well as of virus control; secondly, nAbs and anti-spike IgG persist in the vast majority of recovered patients regardless of disease severity, age and co-morbidities for up to eight months from symptoms onset; thirdly, SARS-CoV-2 infection back-boosts pre-existing antibody responses to seasonal beta coronaviruses without dampening the development of neutralizing or binding SARS-CoV-2 antibodies nor enhancing critical illness.

Recently, Zohar et al. reported a correlation between critical disease outcome and lack of specific functional IgG responses, but not with nAbs in COVID-19 patients[3]. Previous reports raised doubts as to the efficacy of the protection conferred by nAbs in severe COVID-19 and suggested that enhanced nAbs might

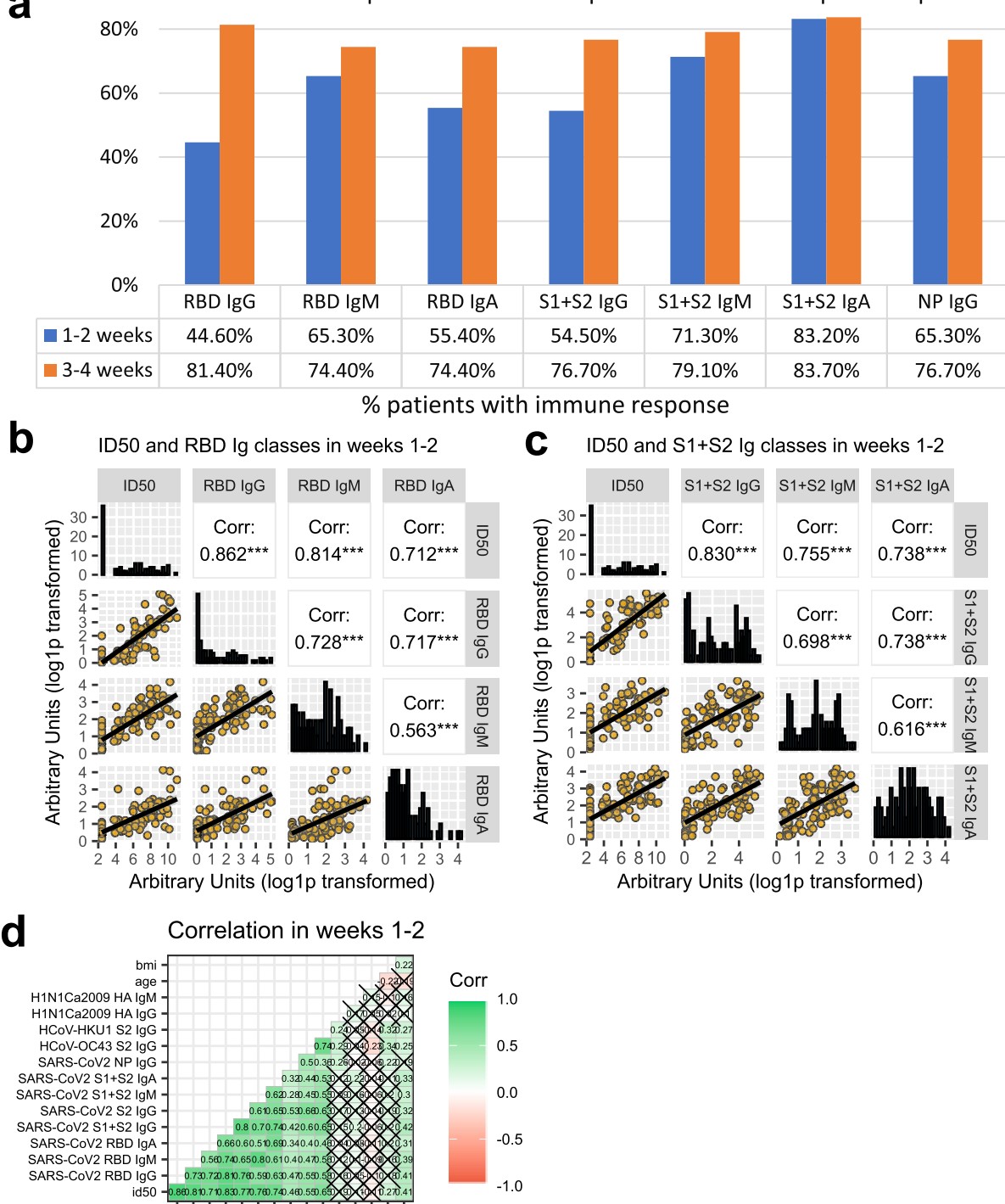

**a** Immune responses of COVID-19 patients at the in-hospital sample

| | RBD IgG | RBD IgM | RBD IgA | S1+S2 IgG | S1+S2 IgM | S1+S2 IgA | NP IgG |
|---|---|---|---|---|---|---|---|
| ■ 1-2 weeks | 44.60% | 65.30% | 55.40% | 54.50% | 71.30% | 83.20% | 65.30% |
| ■ 3-4 weeks | 81.40% | 74.40% | 74.40% | 76.70% | 79.10% | 83.70% | 76.70% |

% patients with immune response

**b** ID50 and RBD Ig classes in weeks 1-2

**c** ID50 and S1+S2 Ig classes in weeks 1-2

**d** Correlation in weeks 1-2

correlate with a negative clinical outcome[12–14]. Conversely, in our study an absent nAb response early after disease onset, more than a difference in titer, was the strongest correlate with both death and delayed viral control. Aware that our patient cohort was characterized by an advanced age and diverse co-morbidities, we evaluated their impact on nAb kinetics. Indeed, the patients that did not develop nAbs were mostly elderly men (median age 74 years, range 59-79) with chronic severe co-morbidities. However, upon controlling for these factors, we confirmed that a compromised immune response to the spike (rather than an

**Fig. 3 Anti-SARS-CoV-2 Spike neutralizing and binding antibody profile. a** Indicated is the percent of 150 COVID-19 patients tested at the in-hospital visit, that had IgG, IgM, and IgA to SARS-CoV-2 antigens detected with LIPS, when sampled during the first or second two weeks from symptoms onset. **b**, **c** Sera of COVID-19 patients, collected at the indicated timepoints from symptoms onset, were measured by LV-based-neutralization assay and the LIPS indicated in gray labels above each row/column. In (**b**) are Ig to RBD and in (**c**) Ig to S1 + S2. Boxes under the diagonal show each correlation plot of the reciprocal of ID50 and arbitrary units after log1p conversion. Dots correspond to individual measurements, the black line represents the regression line and the gray area is 95%CI. Boxes on the diagonal show as histograms the distribution of values in each assay. Boxes above the diagonal show the corresponding Pearson correlation analysis coefficients. Asterisks correspond to the following p values: ***$p \leq 0.001$; **$p \leq 0.01$; *$p \leq 0.05$. **d** Correlation matrix of the indicated variables in weeks 1–2. For each pair, Pearson's correlation coefficient is shown as number and on a color scale. Statistically non-significant correlations are crossed. Source data are provided as a Source Data file.

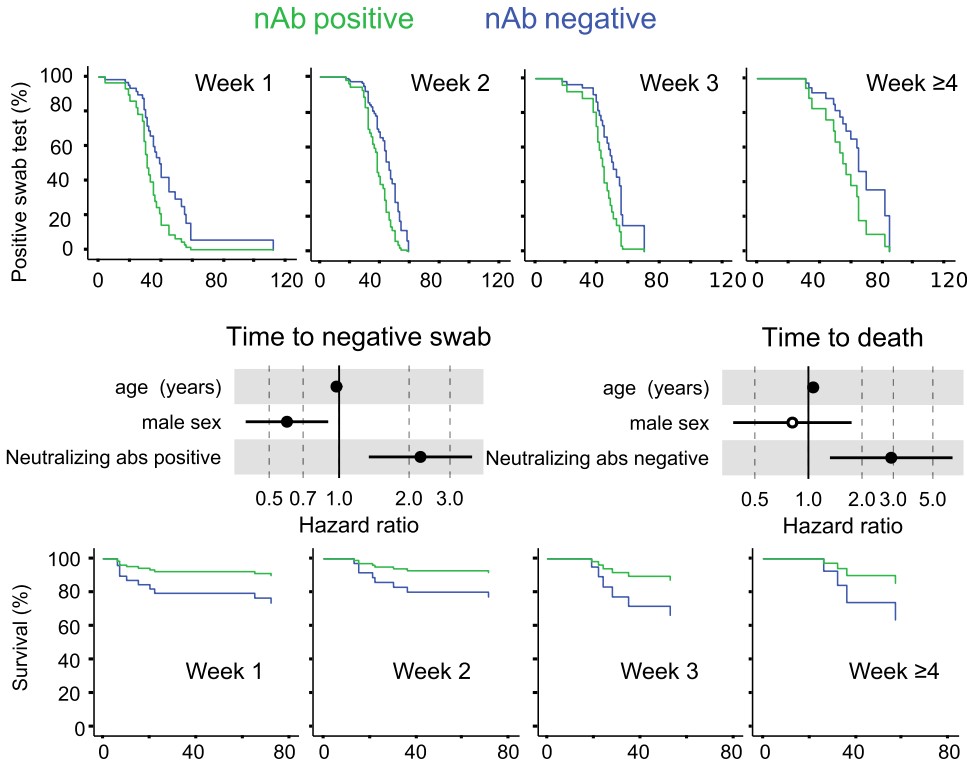

**Fig. 4 Early development of the nAb response correlates with viral control and survival.** Cox regression survival estimates for 150 patients with COVID-19 sampled for nAb response to SARS-CoV-2 during in-hospital visit. The analysis was adjusted for sex and age and stratified for the time from symptoms onset at serum sampling. In (**a**) the time to a SARS-CoV-2 negative naso-pharyngeal swab (HR 2.238, 95%CI 1.342–3.734; $p = 0.002$) and in (**b**) the survival rate (HR 2.918, 95%CI 1.321–6.449; $p = 0.008$) was estimated by the presence or absence of a nAb response. Dots represent the Hazard Ratio (HR), filled dots stand for $p < 0.05$ (two-sided). Wald statistics were used for comparison. Source data are provided as a Source Data file.

enhanced one) is a major trait of patients with critical conditions. Conversely, patients with nAbs and anti-spike IgA showed a faster control of the virus. Although the LIPS assay could not discriminate between IgA1 and IgA2, the correlation we found concerning anti-spike IgA with a faster control of the virus suggests that they were IgA1 or that IgA1 were more abundant than IgA2 since these latter ones have been shown to elicit a neutrophil pro-inflammatory response correlated with fatal outcome in COVID-19 patients[22].

In this report, we used spike-pseudotyped viral particles instead of the live virus, for evaluating viral neutralization. This may pose some limitations, as folding, cleavage and density of spike proteins on the pseudoviral particles may differ from those of the virus[23–26] and may influence the process of entry and/or the aptness of nAbs to bind virions and thus to neutralize them[27,28]. Yet a strong correlation between pseudovirus- and live SARS-CoV-2 virus-based-neutralization assays has been demonstrated[29,30], suggesting that well-standardized assays using spike-pseudotyped viral

particles represent a valuable and safe compromise to measure virus neutralization. Importantly, anti-spike binding antibodies correlated with nAbs detected with our assay, confirming their potential usefulness as proxy biomarkers of a nAb response.

Recent data suggest that a cocktail of virus-specific mAbs was efficient in reducing the viral load in symptomatic out-patients with high viral load but as yet no IgG and IgA response[5]. Similarly, plasma with high titer nAbs administered to elderly patients within 3 days after symptoms onset reduced the risk of progression to severe respiratory disease[6]. Currently, the Food and Drug Administration has granted an emergency use authorization for the virus-specific mAb therapy only for out-patients with mild-moderate COVID-19[31]. Considering that elderly males with chronic co-morbidities are extremely vulnerable to severe COVID-19 in the absence of an anti-spike response, we believe that these patients should be promptly identified and immediately start therapeutic interventions aimed at restoring the lack of a functional humoral immunity.

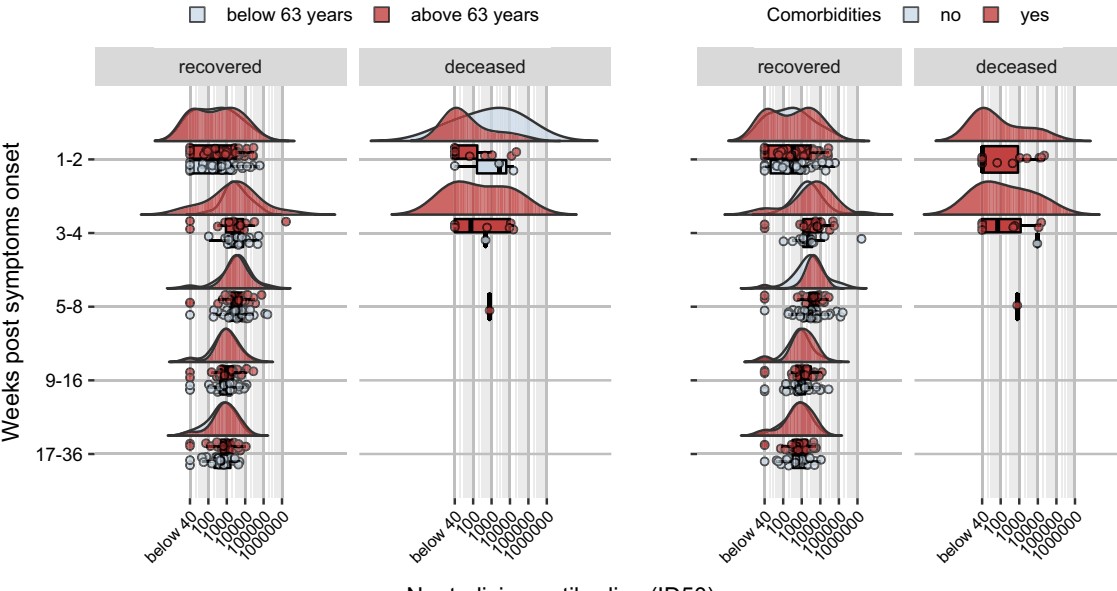

**Fig. 5 Development and kinetics of the nAb response according to age or co-morbidities.** Raincloud plot of each patient's nAb response (circles) expressed as the reciprocal of the ID50 titer. Patients are stratified by time from symptoms onset at sampling (vertical panels, weeks) and clinical outcome (recovered or deceased). Patients are further stratified as below or above the median age of the overall study population (<63 years = blue, >63 years = red) or according to the presence of any co-morbidity (present = red, absent = blue). Shown are the probability density estimate (with the half violin plot upscaled to maximum width for better visualization) and box plots displaying median, IQR, and whiskers extending to 1.96 times the IQR. Source data are provided as a Source Data file.

Interestingly, our data concerning a pre-existing immunity to seasonal HCoVs suggest it not to be detrimental to the development of SARS-CoV-2 responses. Indeed, in contrast with Aydillo et al. and in agreement with Anderson et al.[20,21], who analyzed smaller cohorts over a shorter time span, we found no correlation between the expansion of IgG to HCoVs and a delayed development of SARS-CoV-2 specific antibodies. Our patients exhibited a back-boosting of antibodies to OC43 and HKU1 S2, which is not surprising in light of the structural and partial sequence homology (approximately 40%) between SARS-CoV-2 and HCoVs S2 domains. The correlation of nAb titers with the IgG responses to seasonal beta coronaviruses was good in the first weeks from disease onset, although invariably less pronounced than that with SARS-CoV-2 spike IgG. Importantly, through our LV-based assay we excluded that HCoV-S2 IgG positive, but SARS-CoV-2 spike IgG negative sera could neutralize SARS-CoV-2. Since IgG to HCoV S2 are cross-reactive to SARS-CoV-2 but are likely of low affinity and non-neutralizing the risk of antibody-dependent enhancement cannot be excluded.

As generally expected after the resolution of a viral infection, the nAbs started to drop in recovered patients after 5–8 weeks from symptoms onset. Nevertheless, in all but three patients, nAbs above the assay threshold persisted for as long as eight months. Neither COVID-19 severity at disease onset, nor age, nor the presence of multiple co-morbidities affected the kinetics and persistence of nAbs. At the last follow-up visit, the distribution of nAbs titer was wide and spanning four orders of magnitude starting from close to the assay threshold. However, regardless the presence of long-term detectable antibodies, infection and/or vaccination will induce a memory B-cell response[14,32], that might rapidly mount a specific humoral response. At the third follow-up visit in nine of the 42 nAb-positive COVID-19 patients we documented a major increase of nAb titer (up to 400%) paralleled by levels of IgG to spike antigens (Supplementary Fig. 8). We cannot conclusively state that this rise was due to a re-exposure to

SARS-CoV-2 since none of the patient declared any COVID-related symptoms nor was a virus detection test performed. However, the time of sampling during follow-up of these patients overlapped with the second epidemic wave in Italy[33]. We can speculate that if indeed a re-exposure to the virus occurred, followed by either an asymptomatic or aborted re-infection, then the boost of their antibody levels hints to a responsive adaptive immune system with the ability to control the disease.

While regular re-exposure may sustain protective antibodies, this is based on the prediction that new resistant viral variants will not arise. Although new variants are circulating already worldwide, the impact of the spike/RBD mutations has still to be defined[34–36]. Of note, our LV pseudoparticles bear the Wuhan-01 spike while, at the time of our study, the Italian epidemics were characterized by the predominance, if not exclusiveness, of the SARS-CoV-2 carrying the spike D614G-variant[37]. Nonetheless, our neutralization assay detected high titer nAbs in the majority of patients. This is a partially reassuring evidence, boding well for the vaccination campaign just started in Italy and worldwide with immunogens based on the original Chinese variant. However, should novel strains with nAbs escape mutation(s) arise, a plan of intervention would require updated spike sequences for vaccination. Ongoing vaccination and re-infection studies combined with antibody kinetics to the spike in natural infection will contribute to define correlates of protection and clarify factors that impact on adaptive immune responses, which in turn would be beneficial for modeling the future COVID-19 dynamics[38].

## Methods

**Study population**. We included for the immunological study 162 patients with confirmed SARS-CoV-2 infection enrolled since February 25, 2020 in the COVID-19 clinical-biological cohort study (COVID-BioB) at the IRCCS San Raffaele Hospital (ClinicalTrialsgov identifier NCT04318366). COVID-BioB (protocol number 34/int/2020) and the related immunological sub-study -Role of the immune response in the infection with SARS-CoV-2 and in the pathogenesis of COVID-19-, ImmCOVID, (protocol number 68/INT/2020) were

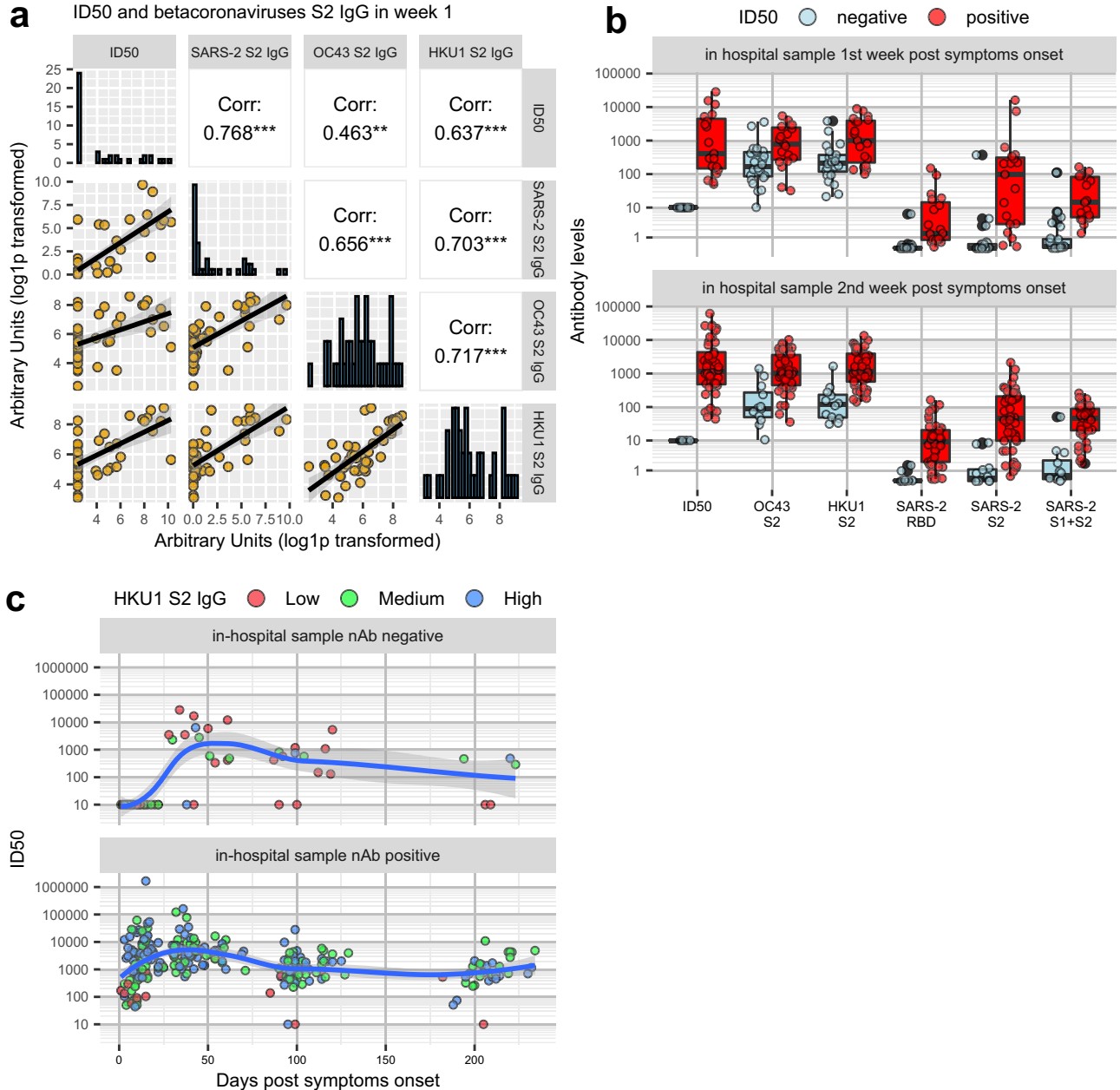

**Fig. 6 NAbs and antibody (IgG) to S2 HCOV-OC43 and HKU1 and SARS-CoV-2.** Sera of COVID-19 patients, collected at the indicated timepoint from symptoms onset, were measured by LV-based-neutralization assay and the LIPS to SARS-CoV-2 and HCoV S2 spike antigen as indicated. **a** Boxes under the diagonal show each correlation plot of the reciprocal of the ID50 and arbitrary units after log1p transformation. Dots correspond to individual measurements, the black line represents the regression line and the gray area its 95%CI. Boxes on the diagonal show as histograms the distribution of values in each assay. Boxes above the diagonal show the corresponding Pearson correlation analysis coefficients. Asterisks correspond to the following *p* values: ***$p \leq 0.001$; **$p \leq 0.01$; *$p \leq 0.05$. **b** Shown are box plot displaying median of each antibody response as indicated, IQR, and whiskers extending to 1.96 times the IQR, stratified by sampling at the indicated week from symptoms onset and a negative or positive nAb response. **c** Development and kinetics of nAb responses to SARS-CoV-2 during follow-up of patients with negative or positive nAb score at the in-hospital sampling. Each colored dot corresponds to the reciprocal of the ID50 of a serum of a given infected individual stratified for low, medium or high IgG to HCoV S2 antigens. Shown is the moving average of data + SE (black curve line + gray band) as obtained by a LOESS curve fitting polynomial regression. Source data are provided as a Source Data file.

reviewed and approved by the Review Board (RB). Patients admitted to the Emergency department (ED) since February 25, 2020 were enrolled according to the protocol described in[15]. All enrolled participants gave written, informed consent[15].

A confirmed infection case was defined as a SARS-CoV-2 positive real-time reverse-transcriptase polymerase chain reaction (RT-PCR) from a nasal-pharyngeal swab and/or symptoms and radiological findings suggestive of COVID-19 pneumonia. Routine blood tests included complete blood count with differential,

renal and liver function tests, C-reactive protein (CRP) and lactate dehydrogenase (LDH), while serum ferritin, D-dimer and interleukin-6 (IL-6) were on request for clinical reasons (Supplementary Table 1). The clinical and laboratory data were collected from medical chart review or directly by patient interview, crosschecked for accuracy by data managers and clinicians and entered in a dedicated electronic case record form (eCRF) developed on site for the COVID-BioB study in MySQL (V5.7.14) on Apache Tomcat (V2.4.23) platform running on windows sever 2012 R2.

Biological material for COVID-BioB included a dedicated serum sample collected at hospital attendance (in-hospital sample) either in the Emergency Department (ED) or the ward and follow-up samples collected in the post-COVID19 outpatient clinic at planned 1, 3 and 6 months visits post-discharge. All serum samples were coded and anonymously processed until disclosure of the COVID-19 status for the immunology study, which was closed by November 25, 2020.

An in-hospital sample was available for 150 patients, while follow-up samples were available for 87, 77, and 46 patients at the month 1, 3, and 6 visits, respectively (Table 1). Twelve patients with no history of hospital admission were followed at the outpatient clinic for up to five visits. A total of 362 sera of COVID-19 confirmed patients were tested for SARS-CoV-2 nAbs response and antibody binding to SARS-CoV-2, HCoV-OC43, HCoV-HKU1, and H1N1 Influenza. Another 17 individuals admitted to the hospital during the same period with respiratory symptoms, but no SARS-CoV-2 diagnosis, were included.

**Plasmids for lentiviral vector pseudotypes expressing luciferase.** Schematic representation of plasmids used in this report is shown in Supplementary Fig. 1. The SIV-based self-inactivating (SIN) lentiviral transfer vector expressing luciferase (Luc) reporter gene (pGAE-LucW), the SIV-based packaging plasmid (pAdSIV3+) producing the proteins necessary for particles formation and the phCMV-VSV.G plasmid producing the pseudotyping vesicular stomatitis virus envelope glyco-protein G (VSV.G) have been already described[39]. For the construction of pSpike plasmid, the full-length codon optimized SARS-CoV-2 spike protein open reading frame (ORF) (GenBank: NC_045512.2) was chemically synthesized, inserted into pUC57 plasmid (TwinHelix, Milan, Italy), excised with AgeI/SalI restriction enzymes and cloned into pEGFP-C3 plasmid (Clontech Mountain View, CA, USA). For the construction of pSpike-C3 plasmid expressing the wild-type spike ORF with a 21 bp deletion at the cytoplasmic tail, the pSpike plasmid was digested with BamHI restriction enzyme and self-ligated.

**Production of lentiviral vector pseudotyped with Spike and expressing luciferase.** Lenti-X human embryonic kidney 293T cell line (Clontech) was maintained in Dulbecco's Modified Eagles (DMEM) High glucose 4.5 g/L (Gibco, Life Technologies Italia, Monza, Italy) supplemented with 10% fetal calf serum (Corning, Mediatech inc. Manassas, VA, USA) and 100 units/mL penicillin/streptomycin (Gibco). For the production of Lentiviral vectors expressing luciferase (LV-Luc), $3.5 \times 10^6$ 293 T Lenti-X cells were seeded on 10 cm Petri dishes (Corning Incorporated - Life Sciences, Oneonta - NY, USA) and transiently transfected with plasmids pGAE-LucW, pADSIV3+ and pseudotyping plasmid (phCMV-VSV.G, pSpike or pSpike-C3) using the JetPrime transfection kit (Polyplus Transfection Illkirch, France) following the manufacture's recommendations using a 1:2:1 ratio (transfer vector: packaging plasmid: envelope/spike plasmid). At 48 h post transfections, culture supernatants containing the LV-Luc pseudotypes (LV-Luc/Spike-C3, LV-Luc/Spike and LV-Luc/VSV.G) were collected, cleared from cellular debris by low-speed centrifugation, passed through a 0.45 μm pore size filter unit (Millipore, Billerica, MA) and stored in 1 mL aliquotes at −80 °C until use. To produce spike-pseudotyped LV-Luc stocks for Western blot analysis, vector containing supernatants were concentrated by ultracentrifugation (Beckman Coulter, Fullerton, CA, USA) on a 20% sucrose cushion (Sigma Chemical Co., St. Louis, MO, USA) at 23.000 rpm for 2.5 h at 4 °C using a SW28 swinging bucket rotor (Beckman). Purified pelleted vector particles were resuspended in 1X phosphate-buffered saline (PBS, Gibco, Life Technologies Italia, Monza, Italy) and stored at −80 °C until use. Each LV stock was measured by the reverse transcriptase (RT) activity assay[40] and titrated for luciferase activity on ACE2 + cells, as described below.

**Flow cytometry to evaluate spike expression.** 293T Lenti-X cells were seeded in 10 cm plates and transfected with 5 μg of pSpike or pSpike-C3 plasmid using the JetPrime transfection kit (Polyplus Transfection). At 48 h post-transfection, 500.000 cells were collected, washed with PBS 1X (Gibco) and fixed with 1% paraformaldehyde. Spike expression was evaluated by flow cytometry with anti-spike S2 rabbit polyclonal Ab (Sino Biological, Prodotti Gianni, Italy, Cat. 40590-T62; used at dilution 1/4000) and PE-conjugated donkey anti-rabbit IgG (Biolegend, San Diego, CA, USA; used at dilution 1/150) using a FACSCalibur flow cytometer (BD Biosciences, Milan, Italy), and data were analyzed by CellQuest Pro software (BD Biosciences). Gating strategy is depicted in Supplementary Fig. 9.

**Western blot.** To evaluate spike presence on LV particles, pellets of concentrated preparations were resuspended and lysed in SDS loading buffer. Lysed virions were separated on 10% SDS polyacrylamide gel under reducing conditions and transferred to a nitrocellulose membrane with a Trans-Blot Turbo System (Bio-Rad, CA, USA). Filters were saturated for 1 h with 5% nonfat dry milk in TSBS (TBS with 0.1% Tween 20) and then incubated with polyclonal anti-spike (Sino Biological, Prodotti Gianni, Italy, Cat. 40590-T62; used at dilution 1/2000) or anti-Gag antibody (NIH Repository Reagents, USA, Cat. #4250; used at dilution 1/400), as primary antibodies, for 2 h at room temperature, followed by incubation for 1 h at room temperature with a goat anti-rabbit horse radish peroxidase (HRP)-conjugated IgG (Bio-Rad, Hercules, USA, Cat. 170-6515; dilution 1/3000). The

immunocomplexes were visualized using the chemiluminescence ECL detection system (WesternBright ECL, Advansta).

**Cell lines.** Caco2 (human, colon), Calu3 (human, lung), HUH7 (human, liver), VEROE6 (African green monkeys, kidney), and 293T (human, kidney) cell lines were grown in Dulbecco's Modified Eagle Medium (DMEM) with Ultraglutamine (EuroClone, Italy) supplemented with 10% (or 20% for Calu3) Fetal Bovine Serum (FBS, EuroClone, Italy), 10 mM of HEPES (1 M) (EuroClone, Italy), 1% Penicillin-Streptomycin (100X) (EuroClone, Italy), 1% Non Essential Aminoacids (NEAA) (100X) (EuroClone, Italy), 1% NA Pyruvate (100 mM) (Euroclone, Italy); Raji (human B cell lymphoma) cells were grown in RPMI 1640 medium with ultra-glutamine (EuroClone, Italy) supplemented with 10% FBS, 10 mM of HEPES (1 M), 1% Penicillin-Streptomycin (100X), 1% NEAA (100X), 1% NA Pyruvate (100 mM). All cell lines were cultured in incubators set at 37 °C with 5% $CO_2$ and passaged every 2–3 days.

**Flow cytometry to evaluate ACE2 expression.** To verify the expression of ACE2 on the cell lines, 200.000 of each cell type were stained with mouse anti-human ACE2 (Millipore, Cat. MAB5676; used at 4 μg/ml) for 30 min at +4 °C followed by Phycoerythrin conjugated goat anti-mouse secondary antibody (SouthernBiotech, USA, Cat. 1030-09; used at dilution 1/300) for 30 min at +4 °C. The samples were acquired with a Navios flow cytometer (Beckman Coulter Inc., Brea, CA, USA) and analyzed using FlowJo version 8.8.7 (Tree Star) with a gating strategy shown in Supplementary Fig. 2.

**Infectivity assay.** Infection of a panel of four epithelial cell lines of human or macaque intestinal or pulmonary origin, and a control B cell line was performed with LV-Luc/Spike and LV-Luc/Spike-C3 pseudoviruses in replicates of three serial dilutions starting from an initial 2-fold dilution of the stock lentiviral preparations. Two or more cell densities were used for each cell line and cultured to test vitality and density after 24 and 48 h. Caco2 (40.000 up to 250.000 cells), Calu3 (50.000 up to 250.000 cells), HUH7 (7.000 or 15.000 cells), VEROE6 (20.000, 50.000 or 100.000 cells), and Raji (70.000, 100.000 or 200.000 cells) were seeded at the indicated number per well in growth medium in 96-well culture plates and incubated at 37 °C 5% $CO_2$ until cells were properly settled. Culture fluid was removed and 100 μL of each pseudovirus dilution added to the cells, and further cultured for 24 and 48 h. Cell growth and density were evaluated by optical microscopy observation and vitality by Trypan blue (Sigma) count. The activity of firefly luciferase was measured with a Luc reporter gene assay system reagent (Bright-Glo, Promega, Madison, Wisconsin, US). Briefly, 100 μL of Bright-Glo was added to each well for a 2 min incubation at room temperature in the dark to allow cell lysis, thereafter 100 μL of cell lysate was transferred to 96-well white plate (Perkin-Elmer, Life Sciences, Shelton, CT) for measurements of RLU using a Mithras luminometer (Berthold, Germany).

**Titration assay.** Titration of the pseudoviruses was performed in VEROE6 cell cultures. LV-Luc/Spike-C3 was titered performing five replicates of serial 2-fold initial dilutions, and control LV-Luc/VSV.G was titered performing 10-fold initial dilutions in duplicate in growth medium in 96-well culture plates. Freshly trypsinized VEROE6 cells were added to each well in growth medium and incubated at 37 °C until cells were settled. Culture supernatant was removed and replaced with 100 μL of each virus dilution and further cultured. After 48 h 100 μL of Bright-Glo was added to each well and incubated for 2 min at room temperature in the dark to allow cell lysis; 100 μL of cell lysate was transferred to 96-well white plate and RLU measured using a Mithras luminometer. Two different LV-Luc/Spike-C3 productions were compared and gave similar results. The virus dilution with 150.000-200.000 RLU, which had a coefficient of variation (CV) below 30%, was selected for the neutralization assay.

**Lentiviral vector-based SARS-CoV-2 neutralization assay.** Pseudoviruses LV-Luc/Spike-C3 and control LV-Luc/VSV.G were used in the neutralization assay. Serum was heat-inactivated by incubation at 56 °C for 30 min. LV-Luc/Spike-C3 supernatant was incubated with four or six 3-fold serial dilutions starting with 1/40 of each heat-inactivated serum, for 30 min at 37 °C in 96-well plates. VEROE6 at a density of 20.000 cells/well were added to flat bottom 96-well plates in growth medium. After 4 h of incubation at 37 °C to allow cells to properly settle to the plate, culture supernatant was discarded, and the virus-serum mixture added in duplicates to the cells. Plates were incubated at 37 °C for 48 h, and thereafter the procedure followed the same steps as for the viral titration assay described above, until the measurement of luminescence in a Mithras instrument.

Serum samples from SARS-CoV-2 infected and uninfected individuals (collected pre-2019 and during the epidemic) were used to set the neutralization assay. Sera from 19 uninfected individuals, tested at 1/20 and 1/40 dilution, gave results in the range of the background RLU levels. Aware that some very low level (between 1/20 and 1/39) neutralization titers would be missed, the 1/40 dilution was chosen as the first suitable serum dilution for the assay, with the intention to save valuable biological specimens. None of the sera from the uninfected individuals, which had detectable IgG to HCoVs OC43 and HKU1 and absent IgG to RBD, showed an inhibition in the neutralization assay, and, thus, excluding an

unspecific inhibition of SARS-CoV-2 by these antibodies. The COVID-19 sera were repeatedly tested to set the intra- and inter-assay variation and thus, define the validity criteria as defined herein.

Virus controls were added to each 96-well test plate and included 5 wells with LV-Luc/Spike-C3 and cells. Additional controls were four 3-fold serial dilutions of two sera with a known SARS-CoV-2 neutralization titer and of a negative serum added in duplicate to each plate. The 50% and 75% inhibitory serum dilution (ID50 and ID75) were calculated with a linear interpolation method using the mean of the duplicate responses[41]. Neutralization was expressed as the reciprocal of the serum dilution giving 50% inhibition of RLU compared to the mean of the virus control wells. An ID50 below 1/40 serum dilution was considered negative and a value of 10 ascribed for statistical analysis.

A test was valid if the percent CV RLUs of the virus control wells was ≤30%, and the value of both positive control sera was within a 2-fold range of the median inter-assay ID50 value. A test sample result was discarded and the serum re-tested when the percent difference between duplicate wells was >30% for sample dilutions that yielded at least 40% inhibition.

To exclude any unspecific inhibition each serum was tested in duplicate at 1/40 dilution with LV-Luc/VSV.G in the neutralization assay, as described above. Serum with >50% VSV inhibition compared to virus control was discarded from any further analysis.

**Fluid-phase luciferase immune precipitation (LIPS) assay**. Antibodies were measured by LIPS assays, which were previously standardized and published[4]. Several recombinant monomeric or multimeric SARS-CoV-2 proteins were produced tagged with a Nanoluciferase reporter (Promega, Madison, Wisconsin, USA): the whole spike glycoprotein S1 + S2 (a derivative of the previously described trimeric S1 + S2 spike protein[42]) or its parts: the S2 protein and the glycoprotein RBD, the nucleocapsid protein (NP), the spike S2 proteins of the HCoV-OC43 and HCoV-HKU1 beta coronaviruses and the hemagglutinin HA1 protein of the 2009 H1N1 pandemic flu virus (Supplementary Fig. 10). Recombinant nanoluciferase-tagged antigens were expressed by transient transfection into Expi293F™ cells (Expi293™ Expression System, Thermo Fisher Scientific Life Technologies, Carlsbad, CA, USA) according to the manufacturer's instructions. Upon harvesting, recombinant proteins were aliquoted and stored frozen at −80 °C.

For antibody measurement by LIPS, the antigen of interest was thawed, diluted in 20 mM Tris Buffer, 150 mM NaCl, 0.5% Tween-20, pH 7.4 (TBST) buffer and adjusted to achieve a luciferase activity corresponding to a final concentration of $4 \times 10^6$ Light Units (LU)/25 μL. Serum was then seeded (5 μL for IgM, 1 μL for IgG or IgA measurements, respectively) into the well of a 96-deep-well plate (Beckman Coulter Inc.) and 25 μL of the diluted antigen preparation was added, followed by a 2 h incubation at room temperature. For IgG antibody measurement, immunocomplexes were then captured with 5 μL of a 50% weight/volume blocked rProtein A slurry (GE Healthcare Europe GmbH, Freiburg, Germany) for 1 h at 4 °C with shaking. Plates were washed 5 times by sequentially dispensing 750 μL/well of TBST, followed by centrifugation at 500 g for 3 min at 4 °C and removal of the wash buffer using a micro-plate washer/dispenser (BioTek Instruments Inc., Winooski, VT, USA). For IgM or IgA antibody measurements, rProtein A was replaced with 5 μL of goat anti-human IgM- or anti-human IgA agarose (Merck Life Sciences, Milano, Italy), respectively.

After washing the resin pellets were transferred to an OptiPlate™ 96-well plate (PerkinElmer, Waltham, MA, USA), 40 μL/well of Nano-Glo® substrate (Promega) were added and the recovered luciferase activity was measured over 2 sec/well in a Berthold Centro XS3 luminometer (Berthold Technologies GmbH & Co. KG, Bad Wildbad, Germany). MikroWin version 5.22 was used for collection of luminometer raw data, which were converted to Arbitrary Units (arb. units) using either a local positive serum as index or serial dilutions of a SARS-COV-2 spike protein antibody positive serum (a kind gift of Prof. Ezio Bonifacio, Dresden, Germany). For antibody titrations, the sera that bound recombinant antigens above the linear range of the assay were serially diluted (1/10, 1/100, 1/1000) in TBST, re-tested until binding fell into the linear range and the calculated arb. units corrected by multiplying for the corresponding dilution factor.

Thresholds for antibody positivity were established upon a QQ plot analysis by selecting arb. units values at which the distribution of calculated arb. units deviated from normality. For ubiquitously present antibody responses like those against the 2009 pandemic flu HA and the HCoV-OC43 and HCoV-HKU1 S2 spike proteins, subjects were binned into terciles as follows: OC43 first tercile 0–270.3 arb. units, second tercile 270.4–1469.7 arb. units, third tercile > 1469.7 arb. units; HKU1 first tercile 0-312.3 arb. units, second tercile 312.4–1471.9 arb. units, and third tercile > 1471.9 arb. units.

**Statistical analysis**. Median with either 95%CI, range, or inter-quartiles range (IQR) and frequency or percent were used to report continuous variables and categorical variables, respectively. Chi-square or Fischer's exact test was used to compare categorical variables. Wilcoxon rank sum or Kruskal-Wallis test were used to compare continuous variables. Imputation for missing data was not performed. Time-to-event was calculated from the date of symptom onset to the date of death, or of last follow-up visit, whichever occurred first. Survival was estimated according to Kaplan–Meier. Association between antibody positivity and time to death was calculated using univariate and multivariate Cox proportional hazards models. The

effect was reported as hazard ratio (HR) with the corresponding 95%CI, estimated using the Wald approximation. All survival analysis and association were stratified according to time from the onset of symptoms to blood sampling (weeks 1, 2, 3, ≥4). Two-tailed p values are reported, with *p*-value < 0.05 indicating statistical significance. All confidence intervals are two-sided and not adjusted for multiple testing. Statistical analyses were performed with SPSS 24 (SPSS Inc./IBM) and the R software version 3.4.3[43].

**Reporting summary**. Further information on research design is available in the Nature Research Reporting Summary linked to this article.

## Data availability statement

The data that support the findings of this study are available from the corresponding author upon reasonable request. Source data are provided with this paper.

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

## Acknowledgements
The work performed at IRCCS Ospedale San Raffaele (OSR) was funded by Program Project COVID-19 OSR-UniSR and Ministero della Salute (COVID-2020-12371617).

The work by Viral Evolution and Transmission Unit, OSR, and Istituto Superiore di Sanità (ISS) was funded by EAVI2020, the European Union's Horizon 2020 research and innovation programme under grant agreement No. 681137. ISS received support in part by NATO multi-year Project No. G5817 "New and Validated Tools for the Diagnosis and follow-up of SARS-CoV-2 Infected Individuals" and by ISS internal funds. We thank Fondation Dormeur, Vaduz for the donation of laboratory instruments relevant to this project to the Viral Evolution and Transmission Unit and ISS. The following reagent was obtained through the NIH AIDS Reagent Program, Division of AIDS, NIAID, NIH: Anti-HIV-1 SF2 p24 Polyclonal. We thank Florian Krammer, Mount Sinai, for kindly donating SARS-CoV-2 Spike plasmids. A special acknowledgement goes to the COVID-BioB team and healthcare workers at OSR that made this work possible (Supplementary Table 5). We thank Priscilla Biswas for English editing and critical revision of the manuscript.

## Author contributions
G.S., V.L., ACara conceived and designed the study. G.S., S.D., V.L. wrote the first draft with contributions from ACara, D.N., L.P., MBaratella. V.L., L.P., G.S. did the statistical analysis. S.D., M.T., F.S., M.S., M.F.P., MBorghi, MLdA, ACannitano, G.V., C.B., E.B., I.M. did the laboratory tests. S.D., MBaratella, V.L., G.S., L.P., C.T. did the data organization, data storage and quality control. C.T., S.D., M.S., M.T. oversaw sample handling and storage. F.C., G.S. oversaw institutional review board submissions, approval, and study oversight. G.S., L.P., V.L., F.C., ACara, D.N. acquired funding. V.L., G.S., D.N., ACara, S.D. prepared the manuscript for submission. These authors contributed equally: S.D, M.S., and M.F.P. These authors jointly supervised this work D.N., ACara, V.L., and G.S. All authors reviewed the manuscript for intellectual content and assisted in data interpretation. All authors confirm that they had full access to all the data in the study and accept responsibility to submit for publication. The sponsors had no role in study design; in the collection, analysis, and interpretation of data; in the writing of the report; and in the decision to submit the paper for publication.

## Competing interests
V.L., M.S., and L.P. have a patent pending that refers to polypeptides, nucleic acids, vectors, and host cells and their use to detect SARS-CoV-2 antibodies with LIPS. All other authors have no competing interests.
