## [Peer Review File · Nature Communications]

REVIEWERS' COMMENTS

Reviewer #1 (Remarks to the Author):

This is a nice study with appropriately drawn conclusions. The results add to a growing body of literature helping to define the role of Ab responses in COVID-19.

The statistical analyses appear to be appropriate.

I have two minor suggestions for the authors to consider that may improve the manuscript:

1. Some copy-editing would be helpful to improve the flow of the writing.
2. With various surrogate assays understandably being used for evaluating clinical samples as opposed to testing with authentic virus, some brief commentary on any limitations the authors may have about their novel assays would be informative.

Reviewer #2 (Remarks to the Author):

The manuscript by Dispinseri et al describes a study directed at understanding the longevity and importance of the antibody response to SARS-CoV-2 in symptomatic COVID-19 patients. The manuscript describes a very thoughtful attempt to define the relative importance of neutralizing antibodies (nAb) responses to infection outcomes. The described study supports the idea that nAb are relevant in the early onset of infection correlating with patient survival further supporting the therapeutic application of monoclonal antibodies for those patients who don't have an adequate immune response. The authors provide detailed relationships that are statistically correlative with various responses and outcomes. Perhaps some more can be said about the IgA response relative to pulmonary symptoms and potential cytokine storm since IgA2 can potentiate such clinically relevant negative inflammatory responses. Overall, the data is presented in a coherent manner and the data supports the conclusions drawn. It is suggested that a grammar and sentence structure check be performed as at times grammar and structure impacts on readability.

Reply to Reviewers.

We thank the reviewers for their comments and suggestions, which we have incorporated in our manuscript. We have edited the manuscript to improve the flow and readability. In specific we have included the following to respond to the reviewer's advise:

Reviewer 1: With various surrogate assays understandably being used for evaluating clinical samples as opposed to testing with authentic virus, some brief commentary on any limitations the authors may have about their novel assays would be informative.

REPLY: we have followed the advise and added the text and references below in the discussion.

“In this report, we used Spike-pseudotyped viral particles instead of the live virus, for evaluating viral neutralization. This may pose some limitations, as folding, cleavage and density of spike proteins on the pseudoviral particles may differ from those of the virus [1–4] and may influence the process of entry and/or the aptness of nAbs to bind virions and thus to neutralize them [5, 6]. Yet a strong correlation between pseudovirus- and live SARS-CoV-2 virus-based neutralization assays has been demonstrated [7-8], suggesting that well-standardized assays using Spike-pseudotyped viral particles represent a valuable and safe compromise to measure virus neutralization. Importantly, anti-Spike binding antibodies correlated with nAbs, confirming their potential usefulness as proxy biomarkers of a nAb response.”

1. Yuste, E., Johnson, W., Pavlakis, G. N. & Desrosiers, R. C. Virion envelope content, infectivity, and neutralization sensitivity of simian immunodeficiency virus. *J. Virol.* 79, 12455–12463 (2005).
2. DeSantis, M. C., Kim, J. H., Song, H., Klasse, P. J. & Cheng, W. Quantitative correlation between infectivity and Gp120 density on HIV-1 virions revealed by optical trapping virometry. *J. Biol. Chem.* 291, 13088–13097 (2016).
3. Stano, A. et al. Dense array of spikes on HIV-1 virion particles. *J. Virol.* 91, e00415–e00417 (2017).
4. Zhu, P. et al. Distribution and three- dimensional structure of AIDS virus envelope spikes. *Nature* 441, 847–852 (2006).
5. Dieterle, M. E. et al. A replication- competent vesicular stomatitis virus for studies of SARS-CoV-2 spike mediated cell entry and its inhibition. *Cell Host Microbe* 28, 486–496 (2020).
6. Case, J. B. et al. Replication- competent vesicular stomatitis virus vaccine vector protects against SARS-CoV-2-mediated pathogenesis in mice. *Cell Host Microbe* 28, 465–474 (2020).
7. Schmidt F, Weisblum Y, Muecksch F, et al. Measuring SARS-CoV-2 neutralizing antibody activity using pseudotyped and chimeric viruses. *J Exp Med.* 2020;217(11):e20201181. doi:10.1084/jem.20201181

8. Sholukh AM, Fiore-Gartland A, Ford ES, et al. Evaluation of SARS-CoV-2 neutralization assays for antibody monitoring in natural infection and vaccine trials. Preprint. medRxiv. 2020;2020.12.07.20245431. Published 2020 Dec 8. doi:10.1101/2020.12.07.20245431

Reviewer 2: Perhaps some more can be said about the IgA response relative to pulmonary symptoms and potential cytokine storm since IgA2 can potentiate such clinically relevant negative inflammatory responses.

REPLY: we have followed the advise and added the following text with the relevant reference in the discussion.

“Conversely, patients with nAbs and anti-Spike IgA showed a faster control of the virus. Although the LIPS assay could not discriminate between IgA1 and IgA2, the correlation we found concerning anti-spike IgA with a faster control of the virus suggests that they were IgA1 or that IgA1 were more abundant than IgA2 since the latter ones have been shown to elicit a neutrophil pro-inflammatory response correlated with fatal outcome in COVID-19 patients (ref :Staats LAN et al, Cells 2020, 9, 2676; doi:10.3390/cells9122676).”